# Resistant Starch Type 2 from Wheat Reduces Postprandial Glycemic Response with Concurrent Alterations in Gut Microbiota Composition

**DOI:** 10.3390/nu13020645

**Published:** 2021-02-17

**Authors:** Riley L. Hughes, William H. Horn, Peter Finnegan, John W. Newman, Maria L. Marco, Nancy L. Keim, Mary E. Kable

**Affiliations:** 1Department of Nutrition, University of California, Davis, CA 95616, USA; rlhughes@ucdavis.edu (R.L.H.); john.newman2@usda.gov (J.W.N.); nancy.keim@usda.gov (N.L.K.); 2Obesity and Metabolism, Western Human Nutrition Research Center, Agricultural Research Service, USDA, Davis, CA 95616, USA; william.horn@usda.gov; 3Department of Food Science & Technology, University of California, Davis, CA 95616, USA; pfinnegan@ucdavis.edu (P.F.); mmarco@ucdavis.edu (M.L.M.); 4Immunity and Disease Prevention, Western Human Nutrition Research Center, Agricultural Research Service, USDA, Davis, CA 95616, USA

**Keywords:** gut microbiota, resistant starch, glycemic response, short-chain fatty acids, wheat, metabolism

## Abstract

The majority of research on the physiological effects of dietary resistant starch type 2 (RS2) has focused on sources derived from high-amylose maize. In this study, we conduct a double-blind, randomized, placebo-controlled, crossover trial investigating the effects of RS2 from wheat on glycemic response, an important indicator of metabolic health, and the gut microbiota. Overall, consumption of RS2-enriched wheat rolls for one week resulted in reduced postprandial glucose and insulin responses relative to conventional wheat when participants were provided with a standard breakfast meal containing the respective treatment rolls (RS2-enriched or conventional wheat). This was accompanied by an increase in the proportions of bacterial taxa *Ruminococcus* and *Gemmiger* in the fecal contents, reflecting the composition in the distal intestine. Additionally, fasting breath hydrogen and methane were increased during RS2-enriched wheat consumption. However, although changes in fecal short-chain fatty acid (SCFA) concentrations were not significant between control and RS-enriched wheat roll consumption, butyrate and total SCFAs were positively correlated with relative abundance of *Faecalibacterium*, *Ruminoccocus*, *Roseburia*, and *Barnesiellaceae*. These effects show that RS2-enriched wheat consumption results in a reduction in postprandial glycemia, altered gut microbial composition, and increased fermentation activity relative to wild-type wheat.

## 1. Introduction

The term dietary fiber includes a wide range of carbohydrates from various sources that cannot be digested by human digestive enzymes and therefore move to the large intestine, where they are fermented by the resident gut microbiota [1,2]. These fiber compounds differ in their physiochemical properties including solubility, viscosity, and fermentability, which have implications for their effects on clinical outcomes as well as their interactions with the gut microbiota [2,3]. Resistant starch is a type of dietary fiber that can be divided into four sub-types [4]. Resistant starch type 2 (RS2) is a naturally-occurring form of starch that is indigestible due to the inaccessibility of its granular structure to digestive enzymes and amylases [4]. Sources of RS2 include high-amylose maize and wheat, raw potatoes, green bananas and some legumes [4,5]. RS2 derived from high-amylose maize has been shown to decrease glycemic response, including postprandial glucose and insulin [6,7,8]. Epidemiological evidence suggests a direct relationship between postprandial glycemia and cardiovascular disease and mortality in individuals both with and without type 2 diabetes [9]. Therefore, RS2 may potentially reduce the risk of the development of chronic conditions such as type 2 diabetes and cardiovascular disease via lowering of postprandial glycemia [10,11]. The demonstration of these effects in clinical trials has garnered qualified health claims from the European Food Safety Authority (EFSA), stating that “replacing digestible starch with resistant starch [from high-amylose maize (RS2)] induces a lower blood glucose rise after a meal” [12] and from the Food and Drug Administration (FDA) that “high-amylose maize resistant starch may reduce the risk of type 2 diabetes” [6]. Therefore, it is of interest to determine whether the effects of RS2 from other sources, such as wheat, may have similar effects on glycemic response.

The gut microbiota is composed of thousands of commensal bacteria that reside primarily in the large intestine but whose effects extend far beyond the gastrointestinal tract, influencing metabolism [13,14], immunity [15,16], and a variety of other functions within the human body. Certain fibers, such as RS2, are not fully digested in the upper digestive tract and therefore provide a substrate for bacterial metabolism in the large intestine. Bacterial fermentation results in the production of metabolites, primarily short-chain fatty acids (SCFAs) including acetate, propionate, and butyrate [1,10]. SCFAs are the most predominant microbial metabolites in the large intestine and impact functions such as glucose homeostasis, inflammation, and satiety [17]. For instance, SCFAs are thought to mediate glucose homeostasis via activation of G protein-coupled receptors (GPR) GPR41 and GPR43, and thereby induce secretion of glucagon-like peptide-1 (GLP-1) and peptide YY (PYY) [17], both of which may also function in satiety [18]. These metabolites form a bridge of communication between microbes and host.

Certain bacterial taxa were shown to be involved in the fermentation of and response to RS2 such as *Ruminococcus bromii*, *Faecalibacterium prausnitzii*, *Bifidobacterium spp.*, *Eubacterium rectale*, *Akkermansia muciniphila*, *Prevotella copri*, and *Bacteroides spp.* [10,19,20]. These microbes may work in combination to ferment RS (primary degraders) and break it down into more accessible metabolites that can then be consumed by other taxa including butyrate producers [10].

A recent study investigated the acute, single exposure effect of high-amylose wheat on glycemic response and found that the glycemic, insulinemic, and incretin responses to high-amylose wheat bread were lower than to low-amylose wheat [21]. However, impacts of regular or consistent supplementation on glycemic response and gut microbiota composition have not been investigated. Therefore, we determined whether RS2-enriched wheat altered the gut microbiota composition and whether these alterations were correlated with metabolic improvements.

## 2. Materials and Methods

### 2.1. Study Design

This trial was conducted in adherence with the Good Clinical Practice guidelines and ethical standards of the Helsinki Declaration. The trial protocol was approved and ethical clearance to conduct this study was granted by the University of California Davis Institutional Review Board (IRB) (Protocol #: 984621). This trial was registered on ClinicalTrials.gov (NCT03082131). Informed consent was obtained from each participant before being enrolled into this study.

Participants were healthy males and females, aged 40–65 years. Participants were excluded if they were outside of the age range, had a body mass index (BMI) < 18.5 or >39.9 kg/m^2^, untreated or uncontrolled metabolic diseases, any gastrointestinal disorders (e.g., Crohn’s disease, irritable bowel syndrome, colitis), cancer or other serious chronic disease, or dietary restrictions that interfered with consuming the intervention foods. Participants were also excluded if they were pregnant, lactating, used tobacco, or any other prescribed or over-the-counter medications that impacted weight loss or metabolism. A total of 128 individuals inquired about this study and completed a telephone interview. Fifty people were determined to be eligible following this interview and were invited for an in-person screening appointment. After this appointment, 12 people were excluded based on lab criteria including fasting glucose > 110 mg/dL, fasting cholesterol > 240 mg/dL, or inability to complete the blood draw. Thirty-seven people were enrolled in this study. From this group, 7 people dropped out of or were excluded from this study before completion, and a total of 30 subjects completed this study (Table 1). Figure 1 shows the Consolidated Standards of Reporting Trials (CONSORT) flow diagram of participants through this study.

Subjects were randomLy assigned to receive either RS2-enriched wheat (RS) first or wild-type wheat (Control) (Figure 2). A computer-generated 4-block randomization scheme stratified by gender was used to assign participants to the respective treatments. RS2-enriched wheat was developed by Arcadia Biosciences (Davis, CA, USA) using TILLING^®^ (Targeting Induced Local Lesions IN Genomes) [5], a non-transgenic (non-genetically modified) technology used to identify desirable genetic traits in a plant population, which can then be increased by natural breeding. The RS2-enriched and wild-type wheat varietals were milled and refined, and roll products were prepared in an Ardent Mills commercial facility with a mix-center and specialty bakery (Ardent Mills, Denver, CO, USA). Products were shipped overnight and stored at −20 °C at the Western Human Nutrition Research Center (WHNRC) metabolic kitchen prior to dispensing to research volunteers.

Rolls made from RS2-enriched wheat and wild-type wheat were provided as a supplement food for participants’ usual diet for seven days. The nutritional content of RS and control rolls is shown in Table 2. Women were asked to eat three rolls per day (a half roll at breakfast and lunch, two at dinner) while men were asked to eat four rolls per day (one at breakfast and lunch, two at dinner). The RS2-enriched rolls provided 14–19 g of resistant starch per day, whereas the wild-type wheat products provided only 2–3 g of resistant starch per day. Thus, when added to the typical fiber intake of the American diet (~15 g), the RS2-enriched rolls increased dietary fiber intake to recommended levels (Appendix A). Subjects kept a log of the products eaten and returned unused products at the week’s end. The meal challenge was scheduled on the 8th day. A two-week washout period separated the treatments. A fecal specimen was collected prior to (Pre-RS and Pre-Control) and at the end (RS and Control) of each treatment, and seven dietary recalls [22] were obtained to document usual dietary intake, as shown in Figure 2.

### 2.2. Meal Challenge and Test Protocol

Metabolic responses to a mixed breakfast meal containing either RS2-enriched wheat or wild-type wheat were evaluated. After one week of either RS2-enriched wheat or wild-type wheat, the Resistant Starch Meal was provided to individuals in the RS arm of this study and the Control Meal was provided to individuals in the control arm of this study (Figure 2). The test protocol was approximately 4 h in duration. The meal challenge consisted of a standard breakfast meal (43 g egg patty, 15 g cheddar cheese, and 42.5 g turkey sausage sandwich served on either RS2-enriched roll or control roll toasted in 14.2 g butter) prepared in the WHNRC metabolic kitchen. The nutritional composition of each test meal is shown in Table 3.

Meals were isocaloric but the RS meal provided 19.7 g of total dietary fiber and 9.6 g of RS while the Control Meal provided only 4.7 g and 1.8 g, respectively. Over the course of the test day, four blood samples were obtained by venipuncture: one while subject was fasting, and three following consumption of the test meal at 1, 2, and 3 h post-meal. Blood was drawn into vacutainers containing sodium citrate/potassium oxalate for plasma glucose analysis and put on ice immediately following draw. Blood was centrifuged and plasma aliquoted into cryotubes. For insulin, blood was drawn into vacutainers and left to clot at room temperature for 30 min before centrifuging and aliquoting serum. Samples were stored at −80 °C for further analyses. Breath samples were taken every 30 min: two while the subject was fasting, and six following consumption of the test meal. During the test day, participants also completed a 24 h dietary recall documenting their dietary intake for the day prior to the test day as well as a questionnaire evaluating their gastrointestinal symptoms. The gastrointestinal symptom questionnaire assessed degree of nausea, bloating, GI rumbling, gas/flatulence, abdominal pain, diarrhea or constipation.

### 2.3. Stool Sample Collection

Fecal samples were delivered to the WHNRC within 24 h of collection prior to each treatment when participants came to pick up their rolls and at the end of each treatment with participants came in for the test day. Participants were provided with collection kits, which included a cooler, ice packs, commode specimen collection system, three empty tubes (one containing RNAlater), zipper plastic bags, pen, and instructions. Participants used scoops attached to the caps of the tubes to aliquot portions of the sample into each of the three tubes, keeping the remaining sample in the commode container. Participants stored the two stool sample tubes not containing RNAlater immediately in the freezer if not transporting directly to the WHNRC while the remaining stool sample in the commode container and the RNAlater tube were kept refrigerated within multiple zipped bags to ensure no cross-contamination occurred. Samples were then transported using the cooler and ice packs to the WHNRC, where they were immediately placed in freezer storage at −80 °C. The remaining sample in the commode container was homogenized using a Stomacher^®^ paddle blender (Seward Model 80 Stomacher; Tekmar Company, Cincinnati, OH, USA) before being divided into aliquots in Cryo-Store^®^ vials (Perfector Scientific^®^, Atascadero, CA, USA) for long-term storage at −80 °C. Aliquots were used for stool analyte profiling, pH, and microbiota analyses.

### 2.4. Bacterial 16S rRNA Gene Sequencing

A total of 200 mg of the frozen stool samples were aliquoted into a 2.5 mL screw tube containing 300 mg of zirconium beads and InhibitEX Buffer. The samples were lysed by shaking twice at 6.5 m/s for one minute each time using a FastPrep-24™ 5G machine (MP Biomedicals LLC, Santa Ana, CA, USA). The DNA was purified using the QIAamp Fast DNA Stool Mini kit (Qiagen, Hilden, Germany) following the manufacturer’s instructions. DNA concentration was measured using a NanoDrop™ 2000/2000c Spectrophotometer (Thermo Fisher Scientific, Waltham, MA, USA). DNA was stored at −20 °C.

A barcoded forward primer 319F (5′-ACTCCTACGGGAGGCAGCAG-3′) and the reverse primer 806R (5′-GGACTACHVGGGTWTCTAAT-3′) were used to amplify a 484 bp region spanning the V3 and V4 regions of the 16S rRNA gene sequence [23]. PCR reactions contained 2 µL of the fecal DNA and were performed using the ExTaq DNA Polymerase PCR Kit (TaKaRa, Shiga, Japan) according to the manufacturer’s protocol. PCR was performed on a Bio-Rad T100^TM^ thermal cycler (Bio-Rad, Hercules, CA, USA) by incubation for 3 min at 94 °C, followed by 27 cycles of 45 s at 94 °C, 1 min at 55 °C, and 30 s at 72 °C, and ending with a final elongation of 10 min at 72 °C. Samples were stored at −20 °C until library preparation.

PCR product concentrations were quantified using a Qubit Fluorometer (Thermo Fisher Scientific, Waltham, MA, USA) and then combined in equimolar quantities in a single pool to achieve 30 ng DNA per sample. The DNA library was constructed from this pool using an Ion Plus Fragment DNA Library Kit (Thermo Fisher Scientific, Waltham, MA, USA) with Platinum^®^ PCR SuperMix High Fidelity reagent (Thermo Fisher Scientific, Waltham, MA, USA), following the manufacturer’s protocol. Cleaning and purification were performed after each step using AMPure XP Beads (Beckman Coulter Life Sciences, Mississauga, ON, Canada) and freshly prepared 70% EtOH. Following final adaptor ligation, the concentration and purity of the library was measured on an Agilent 2100 Bioanalyzer (Agilent, Santa Clara, CA, USA). The library was amplified afterwards with the Ion Plus Fragment DNA Library Kit following the manufacturer’s protocol and the final concentration was measured using a Qubit Fluorometer. The library was diluted to a concentration of 30 pM using Low TE Buffer (Thermo Fisher Scientific, Waltham, MA, USA) and was measured once again on an Agilent 2100 Bioanalyzer. The library was then stored at −20 °C until sequencing. Sequencing of the library was performed on an Ion Chef/S5 system (Thermo Fisher Scientific, Waltham, MA, USA). A volume of 25 μL of the 30 pM library was loaded into the library tube of the machine and was sequenced using a 530 Chip (Thermo Fisher Scientific, Waltham, MA, USA).

### 2.5. 16S rRNA Sequence Analysis

Sequenced data were analyzed using QIIME 2-2020.2 [24]. Forward reads were imported and demultiplexed before denoising using DADA2 [25]. The first 50 bp were trimmed from forward reads due to poor quality and reads were truncated at base position 350, resulting in 300 bp amplicons. Ion Torrent sequencing of 300 bp amplicons yielded an average of 3512 sequences per sample, ranging from 101 to 25,610. one sample with a sampling depth less than 533 (sampling depth of 57) was excluded from the analysis. The Greengenes 16S rRNA gene database were trained using the 319F (5′-ACTCCTACGGGAGGCAGCAG-3′) and 806R (5′-GGACTACHVGGGTWTCTAAT-3′) primers with minimum and maximum lengths set to 300 bp and 500 bp, respectively. After taxonomic classification, mitochondria and chloroplasts as well as singleton taxa were filtered from the dataset. Shannon diversity was calculated for each sample using the diversity function in the vegan package in R [26]. Chao1 richness was calculated using the chao1 function in the fossil package in R [27]. Raw sequencing data are available from Qiita (https://qiita.ucsd.edu/ (accessed on 9 April 2020)), study number 13367, and the European Nucleotide Archive (https://www.ebi.ac.uk/ena/browser/home (accessed on 9 April 2020)), accession number ERP125218.

### 2.6. Blood and Breath Analyte Profiling

Plasma glucose was analyzed on a Clinical Chemistry Analyzer (Cobas Integra^®^ 4000, Roche Diagnostics, Rotkreuz, Switzerland) using an enzyme-linked assay. Plasma insulin was measured by an electrochemiluminescence sandwich immunoassay (Meso Scale Discovery). Homeostatic Model Assessment of Insulin Resistance (HOMA-IR) was calculated as fasting insulin (μIU/mL) × fasting glucose (mg/dL) ÷ 405.

End-expiratory breath samples were collected in a dual-bag system (GaSampler System; QT00830-P, 750 mL Single-Patient Collection Bag and QT00843-P, 400 mL Discard Bag; Quintron Instrument Co, Milwaukee, WI, USA). Breath was analyzed by a gas chromatograph (BreathTracker SC, Quintron Instrument Co, Milwaukee, WI, USA). Outputs included hydrogen, methane, and carbon dioxide. Hydrogen and methane were corrected for carbon dioxide to standardize to alveolar gas levels and reported in parts per million (ppm). Breath was injected as 20 mL samples with gas tight syringes (QT02741, 30 mL Plastic Syringe, Quintron Instrument Co, Milwaukee, WI, USA) through a drying tube filled with indicating Drierite (QT01161-C, Desiccant, BreathPrep™, Quintron Instrument Co, Milwaukee, WI, USA). Duplicate analyses were performed for each sample and averaged. The chromatograph was calibrated every two hours with a standard mixture of hydrogen (150 ppm), methane (75 ppm) and carbon dioxide (6.0%), according to the manufacturer’s instructions (QT07230-G, Calibration Gas, QuinGas-3).

### 2.7. Stool pH Measurement

Stool pH was measured using a semi-micro sealed electrode (Thermo Scientific^TM^ Orion Economy Series pH Combination Electrode, Thermo Fisher Scientific, Waltham, MA, USA) and Beckman Coulter Phi350 pH meter (VWR Cat#BK511038, Beckman Coulter Life Sciences, Mississauga, ON, Canada). The pH meter was calibrated per manual instructions at pH 4, 7 and 10 and left in deionized water when not in use during the procedure. Aliquots of stool samples were thawed and mixed with deionized water in a 1:2 ratio. Tubes were vortexed and centrifuged before immersing the pH probe in the resultant fecal water. Samples were measured in duplicate and averages were used as final values for further analysis.

### 2.8. Stool Short-Chain Fatty Acid Analysis

Short-chain fatty acids (SCFAs) were extracted from ~50 mg of fecal samples using a 1:1 methanol/acetonitrile (*v*/*v*) mixture. Briefly, 50 mg of fecal material were enriched with 5 µL of a solution of deuterated SCFAs containing 100 mM d3-acetate, 10 mM d5 propionate, and 200 µL of 1:1 methanol/acetonitrile (*v*/*v*), and homogenized using GenoGrinder 2010 homogenizer (SPEX Sample Prep; Metuchen, NJ, USA) for 8 min at 1200 rpm. Samples were then centrifuged at 4 °C for 10 min at 10,000 rcf. A 150 µL supernatant subaliquot was taken and filtered at 0.1 µm through 96-well filter plates. A volume of 50 µL of filtered supernatant was combined with 50 µL of 40 µM 15:1n5 methyl ester as an internal standard for a final volume of 100 µL. Samples were stored at −20 °C until analysis.

SCFAs were analyzed by gas chromatography single quadrupole mass spectrometry (GC-MS) and quantified against authentic calibration curves with corrections based on the calculated recovery of d3-acetate. Briefly, 1 µL of sample was injected in split mode with a 1:100 split ratio and residues were separated on a (0.25 mm × 30 mm 0.25 µm DBI-WAX UI column (Agilent; Santa Clara, CA, USA), detected after electron impact ionization by both selected ion monitoring and 50–450 Da mass scanning. SCFAs were quantified against four 7-point calibration curves of selected ion monitoring mode data bracketing all reported data. MS data were analyzed using Agilent MassHunter version B0.08.00 software (Agilent; Santa Clara, CA, USA). Ninety samples were run on an Agilent 6890 GC instrument and 30 samples were run on an Agilent 7890B GC instrument connected to an Agilent 5977B MSD (Agilent; Santa Clara, CA, USA).

### 2.9. Statistical Analyses

All metabolic variables and outcomes were assessed for normality using quantile-quantile (Q-Q) plots. For data containing outliers, sensitivity analyses were conducted to determine the effect of removing outliers. For data demonstrating a change in statistical significance as a result of outlier removal, results of both analyses are reported. Outcomes explored in the sensitivity analysis included glucose (fasting), insulin (fasting, incremental area under the curve (iAUC), peak), and HOMA-IR. For normally distributed data, the linear model described below was used.

The effect of the RS intervention on metabolic outcomes was assessed using a linear model in R, controlling for treatment sequence. Baseline measurements collected at screening, such as weight, BMI, blood pressure, and fasting glucose, were included in the model for their respective variables. The outcome in the regression model was the difference between the outcome value after RS treatment and the value after control treatment (e.g., RS–C). Sequence was coded as −0.5 and +0.5 so the regression intercept gives the mean treatment effect (possibly adjusted for baseline). The baseline value was centered to the mean by subtracting the mean baseline value across all subjects. Violin plots were used to illustrate the distribution of data.

Effect of RS wheat supplementation on overall gut microbiota configuration was tested using adonis in the vegan package in R while accounting for subject-specific effects [26]. Treatment effect on specific gut microbiota taxa was investigated using DESeq2, which conducts differential gene expression analysis by calculating logarithmic fold changes using generalized linear models based on data read as following a negative binomial model distribution [28]. DESeq2 (version 1.26.0) was run in R using the full dataset and accounted for subject and treatment. From this, contrasts could then be used to conduct pairwise comparisons of individual treatments. Pairwise comparisons of time points for both Shannon diversity index and Chao1 richness were performed using linear mixed models, controlling for treatment sequence as a fixed effect and participant as a random effect.

To assess correlations of individual bacterial taxa with indices of fermentation, Pearson’s correlation of the top 20 most abundant taxa glommed at the genus level (phyloseq) and fermentation indices were used. Fermentation indices included stool pH and concentrations of total and individual SCFAs (at all time points) as well as breath hydrogen and methane (after RS and Control treatments only). Correlations were analyzed for taxa at all time points as well as at baseline (prior to RS) only. All of the above statistical analyses were conducted using R version 3.6.1.

To ascertain whether microbial taxa predicted changes in SCFAs during the intervention, unsupervised clustering was first used to group microbial taxa before testing the composite groups’ correlations with total and individual SCFA concentrations. Microbial taxa percent data were grouped using a principal components based variable clustering in JMP Pro v14.0, an implementation of the SAS VARCLUS Procedure. (SAS Institute Inc. https://support.sas.com/documentation/onlinedoc/stat/151/varclus.pdf (accessed on 9 April 2020)). The resulting component clusters were normalized with the Johnson transformation [29]. Normality was confirmed with a Shapiro-Wilk test. Associations between microbial clusters and total and individual SCFA levels were then evaluated using stepwise linear regressions using the minimum Bayesian information criterion (BIC) as the stopping function. The resulting multiple regression model was used to generate predictive models of total and individual SCFA concentrations using microbial clusters, gender, and treatment variables.

## 3. Results

### 3.1. Effects of RS2-Enriched Wheat on Dietary Intake and Glycemic Response

Participants’ dietary intake was assessed during each intervention (Appendix A). Fiber was significantly increased during the RS intervention (*p* < 0.001). However, it is worth noting that, although fiber intake reached adequate amounts, the contribution of whole grains to participants’ fiber intake was below the recommended amount [30] for both interventions. This could be due, at least in part, to the replacement of breads and other grains in participants’ diets by the rolls, both of which were considered to be refined wheat products.

After each one-week intervention, the Resistant Starch Meal or Control Meal was administered on test days and participants’ glycemic response was measured for 3 h postprandial. Compared to control, individuals consuming the RS meal had significantly reduced postprandial glucose (*p* = 0.003) and insulin (*p* < 0.001) iAUC (Figure 3). The average decrease in glucose iAUC among participants with RS meal relative to control was −1236.3 + 2078.8 mg·min/dL, and the average decrease in insulin iAUC was −21,936.94 + 26,546.53 pmol·min/L. Peak glucose and insulin were also significantly decreased after the RS meal (*p*_glu_ = 0.004, *p*_ins_ < 0.001) (Table 4). The majority of peak glucose values during test days consuming the RS meal occurred at fasting whereas peak glucose on test days during which the Control Meal was consumed primarily occurred at 1 h postprandial. However, there were no significant effects of treatment on fasting glucose or insulin or on HOMA-IR (*p* > 0.05).

### 3.2. Effects of RS2-Enriched Wheat Consumption on Gut Microbiota

Gut microbial communities were evaluated in fecal samples collected before and after each intervention (control and RS) for a total of four samples per subject. Both alpha and beta diversity were significantly altered by RS intake. The composition of the fecal microbiota, as evaluated by principal coordinates analysis of the weighted UniFrac metric, was significantly different after RS2 consumption compared to the other time points (*p* < 0.001, adonis) (Figure 4A).

RS intervention also decreased both Shannon diversity and Chao1 richness of the gut microbial community compared to all other time points (*p* < 0.001, linear mixed model) (Figure 4B,C). There were no significant differences between other time points.

Differences in the relative abundance of individual taxa following each time point (Pre-Control, Pre-RS, Control, RS) were analyzed using DESeq2 (Appendix A, Figure 5). No significant differences were found between Pre-RS and Pre-Control samples (*p* > 0.05). Appendix A shows all taxa that were differentially abundant between experimental groups before *p*-value adjustment for multiple comparisons. After *p*-value adjustment, six of the taxa remained significantly different between groups (Figure 5). After correction for multiple hypothesis testing, the RS intervention was associated with an increase in *Ruminococcus* and *Gemmiger* compared to Control and baseline (Pre-RS). *Faecalibacterium*, *Roseburia*, and *Bifidobacterium* were also increased after RS compared to baseline, though these effects were not significant compared to Control. Additionally, Bifidobacterium was increased after Control compared to baseline, suggesting that the bifidogenic effect was not specific to the RS2-enriched wheat.

### 3.3. Effects of RS2-Enriched Wheat on Microbial Metabolites

Markers of microbial activity measured on test days included hydrogen and methane in the breath, indicative of gas production from fermentation, as well as stool pH, indicative of SCFA production from fermentation. Markers of microbial activity in the intestine increased following the RS intervention relative to after the Control diet. Both fasting hydrogen (*p* < 0.001) and methane (*p* = 0.03) were higher prior to the test meal during the RS2 wheat intervention (Figure 6). Additionally, breath hydrogen (*p* < 0.001) and methane (*p* = 0.03) AUC measured after the test meal were significantly higher following the RS intervention relative to after the Control intervention (Appendix A).

Quantification of fecal SCFAs showed that acetate was the most abundant, followed by butyrate, and propionate (Appendix A). No significant differences were detected in absolute (pmol/mg) concentrations of fecal SCFAs (Table 5).

Additionally, no significant differences in SCFA concentrations were found between Pre-RS and RS, Pre-Control and Control, or Control and RS fecal samples. There was also no significant effect of treatment on stool pH (*p* = 0.45).

Pearson’s correlation showed a significant positive correlation between fasting breath hydrogen and both absolute and relative concentrations of butyrate (*r*_abs_ = 0.28, *p*_abs_ = 0.03; *r*_rel_ = 0.26, *p*_rel_ = 0.05). Additionally, breath hydrogen AUC was positively correlated with absolute concentration of butyrate (*R* = 0.35, *p* = 0.01) and negatively correlated with relative concentration of acetate (*r* = −0.3, *p* = 0.02).

### 3.4. Relationship between Gut Microbial Composition and Metabolic Indices of Intestinal Fermentation

To examine whether the variability in metabolic response to RS2-enriched wheat consumption could be related to an individual’s gut microbial composition prior to consumption of the RS2-enriched wheat rolls, the proportion of the top 20 most abundant microbial taxa present at baseline (Pre-RS) was examined relative to indices of intestinal fermentation by Pearson’s correlation. These indices of fermentation included breath hydrogen and methane, stool pH, total SCFAs, and both relative percent and absolute concentrations of individual SCFAs. A heatmap of these correlations is shown in Figure 7. Baseline proportions of *Clostridiales* and *Rikenellaceae* were positively correlated with methane production after RS intervention, while Bacteroides showed a strong negative correlation. *Parabacteroides*, *Barnesiellaceae*, and *Blautia* were positively correlated with stool pH after the RS intervention, perhaps indicating a negative effect on SCFA production. Indeed, *Parabacteroides* was negatively correlated with total SCFAs, acetate, and butyrate. Baseline *Alistipes* was also negatively correlated with acetate, but showed a positive correlation with butyrate while baseline *Lachnospiraceae* was positively correlated with propionate following the RS intervention.

When examining the top 20 taxa from all time points (Figure 8), *Gemmiger*, *Faecalibacterium*, *Roseburia*, and *Ruminococcus* (of family *Ruminococcaceae*), which were increased by the RS intervention, showed positive associations with methane production and SCFAs, particularly butyrate, though *Faecalibacterium* was negatively correlated with acetate.

The gut microbial community is complex and individual associations likely do not capture the complexity of interactions in the gut. Therefore, to further examine whether groups of taxa might be more significantly associated with SCFA production variable clustering was performed. Variable clustering of microbial taxa percent composition data resulted in 21 discrete clusters (Appendix A). If considering only microbial taxa, Cluster 13 and Cluster 15 yielded the strongest predictive model of total SCFAs, (root mean square error (RMSE) = 8.4; *r*^2^ = 0.23; *p* < 0.0001). If sex and treatment are allowed into the variable set, both sex and Cluster 2 were also included in the model (RMSE = 7.91; *r*^2^ = 0.34; *p* < 0.0001). The microbial composition of Clusters 2, 13, and 15 are shown in Table 6. The most representative species were *f_Victivallacea*, *g_Butyricicoccus*, and *g_Roseburia* for Clusters 2, 13, and 15, respectively.

The model for total SCFA was defined by the following equation:Total SCFA = [−0.9(Clust 2)] + [−2.2(Clust 13)] + [3(Clust 15] + [−3(Female) or 3(Male)] + 29

Using the same procedure to assess associations with acetate, propionate and butyrate showed similar results (Table 7; Appendix A), with butyrate showing the highest explained variance (34%). The adjusted variable regressions for butyrate are shown in Appendix A. Cluster 15, defined primarily by *g_Roseburia* and *g_Ruminococcus*, was the only microbial cluster that was positively correlated with butyrate while Clusters 2 and 13 showed the opposite trend. These data suggest that fermentation indices are correlated with abundance of individual microbes. The associations between relative abundance of Parabacteroides, increased stool pH, and decreased butyrate concentration were consistent when examined as predictive relationships (abundance at baseline relative to intervention outcome) and as direct associations. Similarly, Bacteroides abundance was negatively associated with methane production both at baseline and throughout the intervention. Therefore, these two taxa are likely strong drivers of individual response to RS2 consumption. Additionally, clusters of microbes, which may be more representative of the cross-feeding interactions that the microbial community uses to ferment RS2, show correlations with the fermentation metabolite, butyrate. Specifically, *g_Roseburia* and *g_Ruminococcus*, as well as associated microbes, are positively associated with butyrate production while microbial clusters associated with *f_Victivallaceae*, *c_Alphaproteobacteria*, *g_Butyricicoccus*, and *g_Coprobacillus* may be negatively correlated with butyrate.

### 3.5. Relationship between Indices of Fermentation and Postprandial Glucose and Insulin

There was a high degree of interindividual variability in metabolic response to the RS and Control treatments. Intraclass correlation coefficients (ICCs) were calculated for glucose and insulin to quantify the degree of heterogeneity between subjects. ICC for glucose was 50.85%, suggesting that 50.85% of variability was due to variability between subjects. Similarly, the ICC for insulin was 47.20%.

Because there is some existing evidence of a relationship between SCFA concentration and insulin sensitivity [31], and because we found strong relationships between the relative abundance of individual taxa and clusters of taxa with butyrate production, Pearson’s correlation analyses were conducted to determine whether fermentation indices, specifically SCFAs, were correlated with individual glycemic response. Analyses showed significant positive correlations between relative propionate and fasting glucose (r = 0.3, *p* = 0.01), relative butyrate and glucose iAUC (r = 0.28, *p* = 0.03) and absolute butyrate concentrations and insulin iAUC (*r* = 0.27, *p* = 0.05) (Figure 9). After correction for multiple hypothesis testing, only the correlation between relative propionate and fasting glucose remained significant (*q* = 0.03).

## 4. Discussion

We found that diets containing adequate fiber with at least 30% of fiber from RS2-enriched wheat lowers postprandial glycemia. These findings are in accordance with previous analysis investigating the metabolic effects of RS2 derived from wheat [21]. The current analysis also expands upon previous findings by showing correlations of the effects of RS2-enriched wheat on the gut microbiota composition and suggests that RS2-enriched wheat is associated with higher proportions of *Ruminococcus* and *Gemmiger*. Additionally, associations of microbial metabolites with glycemic response and related metabolic indices suggest a role of the gut microbiota in mediating and potentially modifying response to RS2-enriched wheat.

The reduction in postprandial glycemia and insulinemia suggests that replacing regular wheat with RS2-enriched wheat may help prevent and potentially manage conditions such as type 2 diabetes by controlling rises in glucose and insulin following ingestion of carbohydrates. Over time, lower circulating insulin leads to upregulation of insulin receptors and increased tissue insulin sensitivity [32]. The lowering of postprandial glucose and insulin was most likely due to the reduced availability of carbohydrates in the RS2 wheat rolls compared to the regular wheat control rolls (9.84 g versus 2.35 g total dietary fiber; 4.78 g vs. 0.92 g RS). While some of the RS may have been created as a result of retrogradation of starch after baking, resulting in the formation of RS3, the majority of the RS in the RS-enriched wheat rolls prior to baking was RS2 and therefore inaccessible due to its crystalline structure. The lower postprandial glucose and insulin responses to the RS2-enriched wheat are consistent with maize-derived RS as well as other types of high-amylose food sources and show similar magnitudes of effect [33,34,35].

The current analysis found that RS2-enriched wheat was associated with a decrease in alpha diversity and increases in starch-degrading bacteria such as *Bifidobacterium* as well as increases in *Ruminococcus*, *Roseburia*, *Faecalibacterium*, bacterial genera known to produce butyrate [36]. Previous studies investigating the effects of RS2 on the gut microbiota have demonstrated similar effects on gut microbiota composition and configuration [19]. Consumption of regular wheat also increased the relative proportion of Bifidobacterium suggesting that the bifidogenic effects of wheat were greater than the effects of RS2. Furthermore, we did not observe significant effects of RS2-enriched wheat on taxa such as *Prevotella*, *Eubacterium*, and *Bacteroides* that have been shown to be involved in RS degradation [10,37,38]. The decrease in bacterial diversity often observed in response to RS intake is presumably due to the enrichment of specific taxa able to efficiently access and metabolize its starch components and/or the byproducts of fermentation by primary degraders [19]. Although higher α-diversity is generally thought to be beneficial, this is not always the case if it is also associated with increased gastrointestinal transit time, which is associated with increased proteolysis and circulation of metabolites of proteolytic catabolism [39].

Variation in findings regarding the changes in relative proportions of specific taxa in response to RS intervention may be due to variation in the gut microbial community composition of individual subjects. Therefore, we examined correlations between individual taxa as well as clusters of associated microbes that may play a role in mediating interindividual variability in response to the RS intervention. Indeed, previous studies of RS supplementation have demonstrated that taxa enriched in “responders” and “non-responders” to RS interventions, *Prevotella copri* and *Bacteroides thetaiotaomicron*, respectively, replicate the observed metabolic responses when transplanted into germ-free mice [40]. Total and individual SCFAs showed the greatest number of associations with relative abundance of bacterial taxa. Individual taxa including *Faecalibacterium*, *Roseburia*, and *Barnesiellaceae* as well as a cluster of associated microbes defined by *Roseburia* and *Ruminococcus* showed positive correlations with SCFAs, particularly butyrate. The enrichment of specific taxa, including *Faecalibacterium* and *Bifidobacterium*, and associated metabolic changes such as production of SCFAs have been shown to improve metabolic indices of type 2 diabetes [41]. Research suggests that *Bifidobacterium* and *Ruminococcus* act as primary degraders of RS2 as evidenced by in vitro growth on RS2 and presence of genes and structures that allow these bacteria to adhere to and degrade RS2, such as amylases and glucan-branching enzymes [38]. By direct fermentation of RS2 or cross-feeding, *Faecalibacterium*, *Roseburia*, and *Ruminococcus* have been associated with butyrate production [42,43], which has many potential benefits including improvement of insulin sensitivity [31]. Additionally, *Bacteroides* and *Parabacteroides* seem to be negatively associated with markers of fermentation, suggesting that these taxa may displace more efficient RS2-fermenting taxa. This finding echoes previous results of the potential role of *Bacteroides* in non-responders to RS intake [40]. Our knowledge of such keystone taxa involved in mediating metabolic responses to dietary components such as RS2-enriched wheat is currently limited. Therefore, future studies should examine whether interindividual differences in metabolic response are correlated with baseline or changes in certain taxa.

Limitations of the current study include its duration, sample type and sample size, and lack of functional and mechanistic testing methodology of the gut microbiota. The duration of this study was too short to determine long-term impacts of the observed short-term changes in postprandial glycemic response on outcomes such as development of type 2 diabetes, weight gain, and cardiovascular disease. Additionally, the relatively short duration of the postprandial testing period (three hours), precluded adequate analysis of the effects of colonic fermentation as it takes approximately six to eight hours for ingesta to reach the large intestine [44]. However, by having participants ingest the rolls for seven days prior to the test day, including dinner the night before the morning test, this essentially “primed the system” [45,46]. The lack of additional postprandial blood samples drawn within the first hour after the test meal is also a limitation and may explain the lack of a glucose peak following the meal. However, analysis of the postprandial samples obtained still provides valuable information about glycemia and insulinemia. Additionally, the use of fecal SCFA measurements, rather than circulating plasma concentrations, may explain the counterintuitive positive correlation between butyrate and postprandial glucose. Evidence suggests that circulating SCFAs are more directly linked to outcomes such as insulin sensitivity and GLP-1 concentrations compared to fecal SCFAs, which are excreted and therefore not absorbed or utilized [47]. The sample size of the current study also limited the ability to reliably utilize complex machine learning models such as Random Forests and other predictive algorithms that have been used to assess the role of the gut microbiota in metabolic responsiveness to dietary interventions [48]. The use of 16S rRNA sequencing used in the current analysis allows for high-level overview of the composition of the gut microbiota. However, methods such as metagenomics and metatranscriptomics would provide strain-level resolution and detection of functional genes and gene expression, which would allow for more accurate classification of taxa involved in RS2-enriched wheat degradation and a more sensitive analysis of variability between individual in both taxonomic and functional differences [49]. Lastly, while human clinical trials are desirable and necessary to assess the effects of dietary components on metabolic health, future analyses using animal and/or in vitro models would help complement the current study to elucidate the mechanisms by which the gut microbiota mediates the metabolic effects of RS wheat and whether transplantation of taxa is sufficient to reproduce these effects. This approach will allow for a comprehensive understanding of the effects of RS2-enriched wheat on the gut microbiota and metabolic health and the development of potential therapeutic solutions to help improve metabolic response to RS2-enriched wheat.

## Figures and Tables

**Figure 1 nutrients-13-00645-f001:**
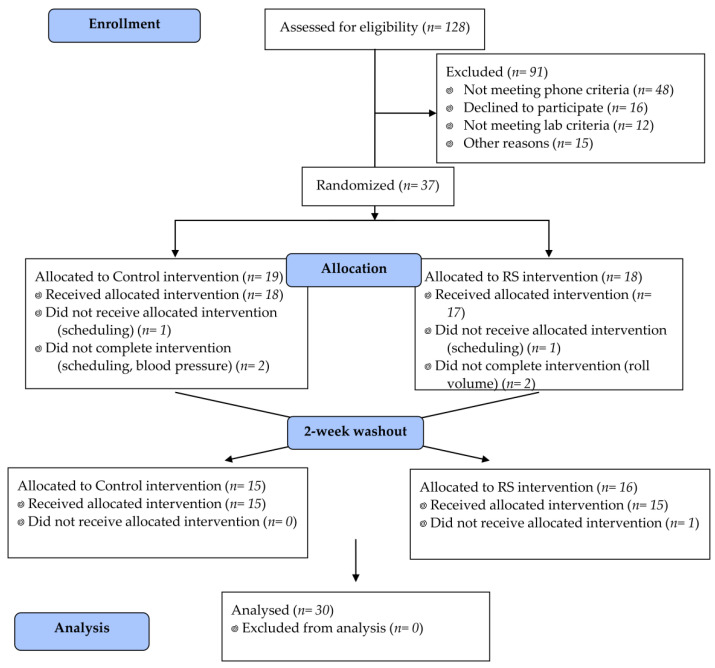
CONSORT diagram. Participant flow in CONSORT-recommended format. CONSORT (Consolidated Standards of Reporting Trials); RS (resistant starch).

**Figure 2 nutrients-13-00645-f002:**
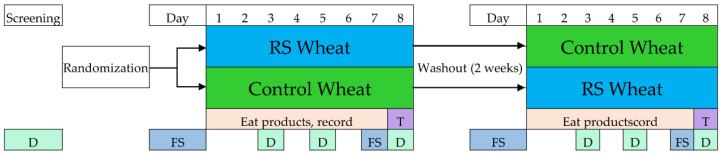
Study design and timeline. RS (resistant starch); T (meal challenge test day); D (dietary recall); FS (fecal sample collection window).

**Figure 3 nutrients-13-00645-f003:**
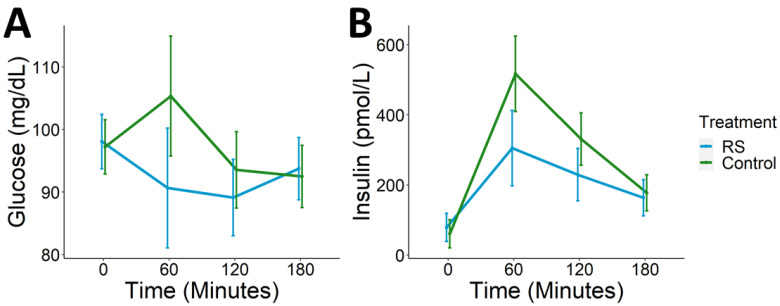
Postprandial glycemic response. RS wheat supplementation was associated with lower postprandial (**A**) glucose and (**B**) insulin (*p*_glu_ = 0.003, *p*_ins_ < 0.001).

**Figure 4 nutrients-13-00645-f004:**
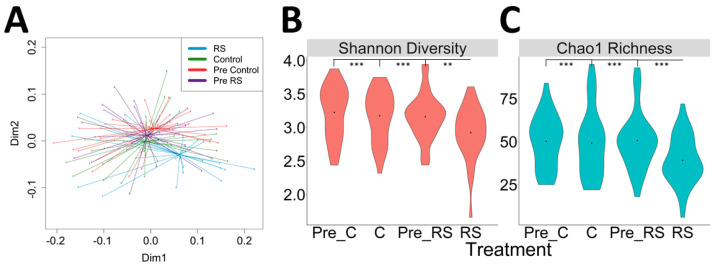
Weighted UniFrac Ordination and Diversity. (**A**) Ordination using weighted UniFrac distances shows significant separate of RS from all other time points (Pre-Control, Control, Pre-RS). (**B**) Shannon diversity was significantly lower after the RS intervention compared to all other time points (**C**) Chao1 richness was significantly lower compared to all other time points. Violin plots shown depict the distribution density with a point at the median. Significance: ** ≤ 0.01; *** ≤ 0.001.

**Figure 5 nutrients-13-00645-f005:**
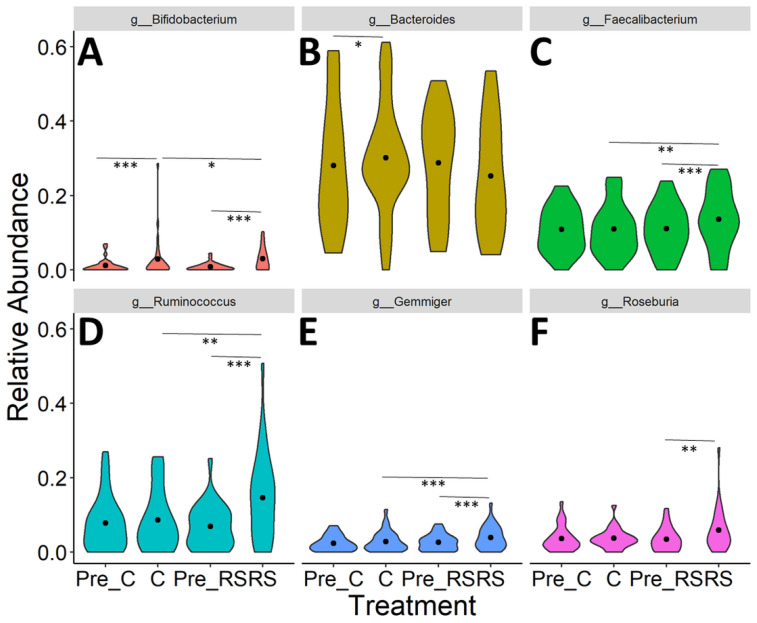
Significant effects of treatment on microbial taxa. DESeq2 identified changes in microbial taxa during RS and control treatments including (**A**) *Bifidobacterium*, (**B**) *Bacteroides*, (**C**) *Faecalibacterium*, (**D**) *Ruminococcus*, (**E**) *Gemminger*, and (**F**) *Roseburia*. Violin plots shown depict the distribution density with a point at the median. Significance: * ≤ 0.05; ** ≤ 0.01; *** ≤ 0.001.

**Figure 6 nutrients-13-00645-f006:**
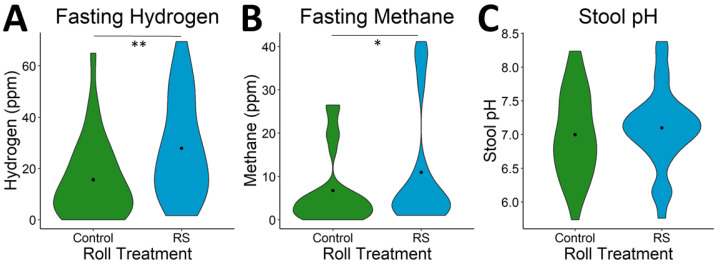
Fermentation response to RS wheat. Fasting (**A**) hydrogen (*p* = 0.001) and (**B**) methane (*p* = 0.03) were increased prior to the test meal after RS wheat supplementation. (**C**) Stool pH showed no significant difference between RS and control. Violin plots shown depict the distribution density with a point at the median. Significance: * ≤ 0.05; ** ≤ 0.01.

**Figure 7 nutrients-13-00645-f007:**
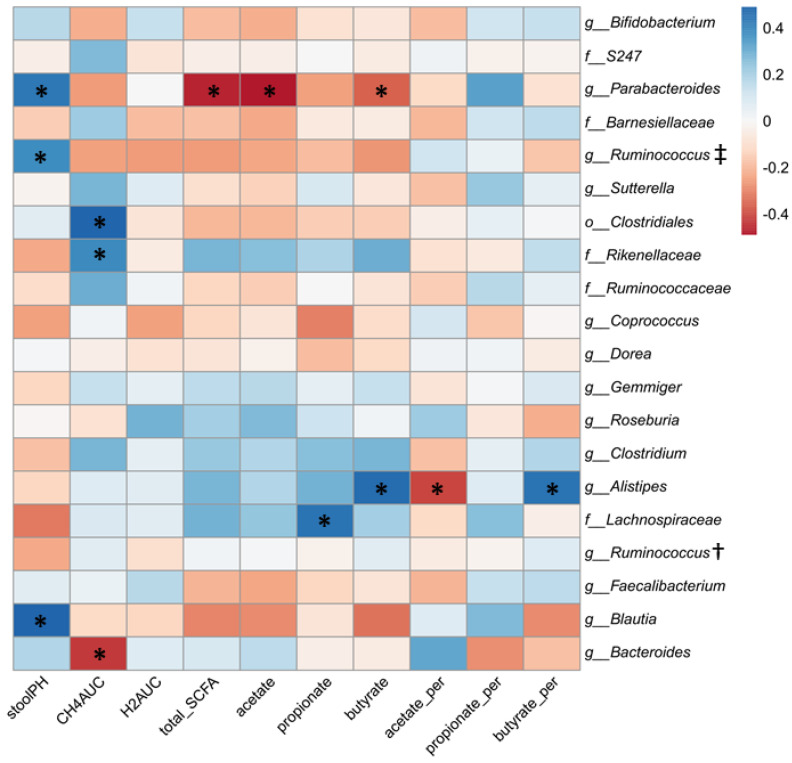
Correlation of top 20 microbial taxa prior to RS with fermentation indices after RS. Pearson’s correlation coefficient (r) values are shown. * Asterisks indicate significant (*p* < 0.05) correlations. † *g_Ruminococcus* of family *Ruminococcaceae*. ‡ *g_Ruminococcus* of family *Lachnospiraceae*.

**Figure 8 nutrients-13-00645-f008:**
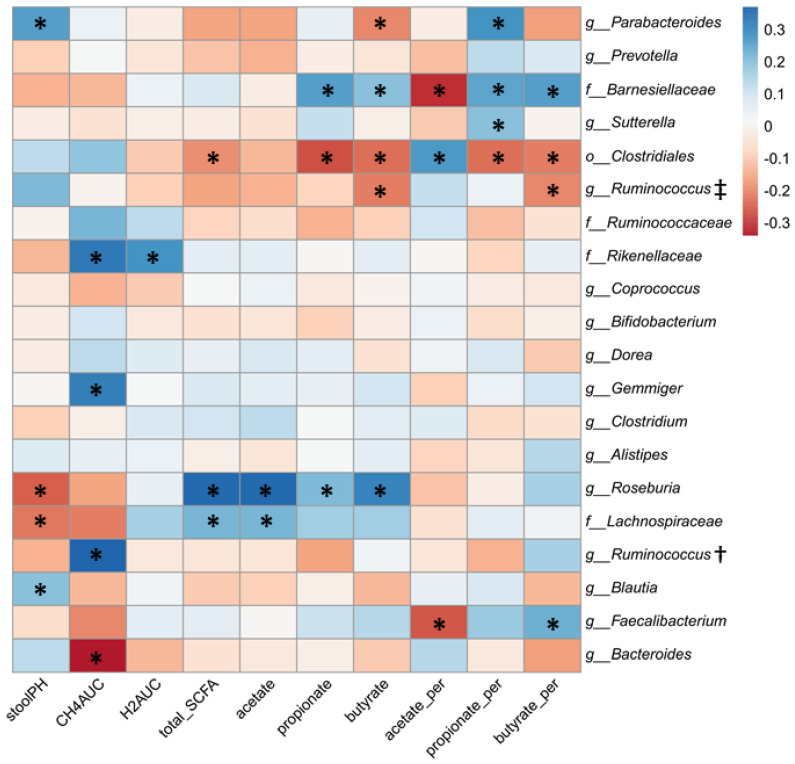
Correlation of top 20 microbial taxa at all time points with fermentation indices from the corresponding time point. Pearson’s correlation coefficient (*r*) values are shown. * Asterisks indicate significant (*p* < 0.05) correlations. † *g_Ruminococcus* of family *Ruminococcaceae*. ‡ *g_Ruminococcus* of family *Lachnospiraceae*.

**Figure 9 nutrients-13-00645-f009:**
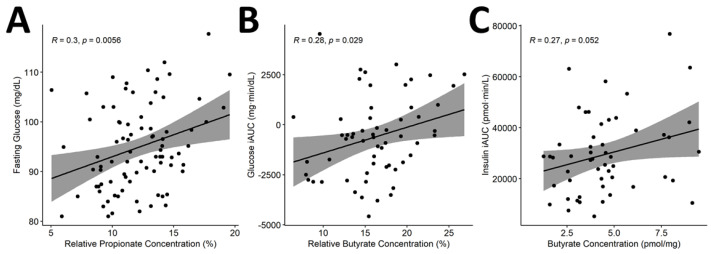
Correlation of SCFAs with glycemic response. Pearson’s correlation showed significant correlations between (**A**) propionate and fasting glucose as well as between butyrate concentrations and (**B**) glucose iAUC and (**C**) insulin iAUC. Relative concentration defined as proportion of total SCFA concentration. Abbreviations: incremental area under the curve (iAUC); short-chain fatty acids (SCFAs).

**Table 1 nutrients-13-00645-t001:** Participant Characteristics.

Participant Characteristic	Male (*n* = 12)	Female (*n* = 18)	Overall Study (*n* = 30)
Age ($\bar{x}$ + sd), years	54.9 (7.6)	53.2 (5.7)	53.9 (6.6)
BMI ($\bar{x}$ + sd), kg/m^2^	26.6 (3.0)	26.0 (4.2)	26.5 (3.8)
W:H ratio ($\bar{x}$ + sd)	0.9 (0.05)	0.8 (0.05)	0.8 (0.1)
Fasting Glucose ($\bar{x}$ + sd), mg/dL	90.8 (6.6)	90.9 (7.6)	90.8 (7.3)
Total cholesterol ($\bar{x}$ + sd), mg/dL	193.2 (17.7)	193.3 (27.2)	193.2 (24.3)
Triglycerides ($\bar{x}$ + sd), mg/dL	122.4 (49.5)	75.2 (26.8)	94.1 (44.8)

Abbreviations: body mass index (BMI); waist:hip ratio (W:H ratio). Values shown are mean ($\bar{x}$) ± standard deviation.

**Table 2 nutrients-13-00645-t002:** Roll nutrition.

Nutrient	Resistant Starch (RS) Roll	Control Roll
Calories (kcal)	242.0	246.6
Carbohydrate (g) *	39.1	44.0
Total dietary fiber (g) ^+^	9.8	2.4
Resistant starch (g) ^+^	4.8	0.9
Insoluble fiber (g) ^+^	5.9	1.1
Soluble fiber (g) ^+^	3.9	1.2
Protein (g)	12.0	10.6
Fat (g)	3.8	3.1
Monounsaturated (g)	1.8	1.6
Polyunsaturated (g)	1.4	1.1
Saturated (g)	0.5	0.4

Footnote: Nutritional content shown is per 1 roll. Average weight of rolls was 92 g. Nutrition information was analyzed by Medallion Labs. * Total carbohydrate was determined by difference and is therefore an estimate; it includes dietary fiber, digestible sugars, and other unmeasured carbohydrates. ^+^ Total dietary fiber represents the sum of insoluble and soluble fiber measured by CODEX (AOAC 2009.01) and gravimetric HPLC (AOAC 2011.25), respectively. Resistant starch measured separately using the Medallion resistant starch test (AOAC 2002.02) and may or may not be included in the insoluble and soluble fiber portions.

**Table 3 nutrients-13-00645-t003:** Test meal nutrition.

Nutrient	Resistant Starch (RS) Meal	Control Meal
Calories (kcal)	780.1	789.3
Carbohydrate (g) *	79.2	88.9
Total dietary fiber (g)	19.7	4.7
Resistant starch (g)	9.6	1.8
Insoluble fiber (g)	11.8	2.3
Soluble fiber (g)	7.8	2.4
Protein (g)	38.0	35.2
Fat (g)	33.7	32.3
Monounsaturated (g)	11.6	11.2
Polyunsaturated (g)	5.9	5.3
Saturated (g)	13.4	13.2

Footnote: Nutrition information for meal components (excluding the rolls) was calculated using Nutrition Data System for Research (NDSR) software, which was then added to the custom nutrition information for the RS2-enriched roll and wild-type wheat roll in the RS meal and Control Meal, respectively. * Total carbohydrate was determined by difference and is therefore an estimate; includes dietary fiber, digestible sugars, and other unmeasured carbohydrates.

**Table 4 nutrients-13-00645-t004:** Postprandial glycemic response.

Glycemic Response	Resistant Starch (RS)	Control
iAUC		
Glucose (mg·min/dL)	−1111.8 ± 1846.3	124.5 ± 2355.3
Insulin (pmol·min/L)	25,147.3 ± 20,486.7	47,084.2 ± 33,989.3
Peak		
Glucose (mg/dL)	104.0 ± 14.5	114.1 ± 24.3
Insulin (pmol/L)	315.6 ± 205.7	532.0 ± 375.7

Footnote: Values shown are the mean ± standard deviation. Abbreviations: incremental area under the curve (iAUC).

**Table 5 nutrients-13-00645-t005:** Concentrations of SCFAs.

		*p*-Values			
SCFA	Concentration (pmol/mg)	Control vs. RS	Control vs. Pre-C	RS vs. Pre-RS	Pre-RS vs. Pre-C
Total SCFAs	28.2 ± 9.6	0.98	0.63	0.37	0.70
Acetate	20.2 ± 6.6	0.93	0.98	0.41	0.35
Butyrate	4.7 ± 2.5	0.58	0.25	0.65	0.21
Propionate	3.4 ± 1.4	0.72	0.18	0.23	0.81
	**Relative Concentration (%)**				
Acetate	72.2 ± 6.7	0.65	0.10	0.20	0.42
Butyrate	15.9 ± 4.8	0.29	0.23	0.64	0.07
Propionate	11.9 ± 3.4	0.61	0.12	0.07	0.43

Absolute (pmol/mg) and relative (%) concentrations of SCFAs acetate, propionate, and butyrate are shown as the mean ± standard deviation. No significant differences between treatments were detected for absolute or relative concentrations. Abbreviations: Resistant Starch (RS), before-Control (Pre-C), before-Resistant Starch (Pre-RS), short-chain fatty acids (SCFAs).

**Table 6 nutrients-13-00645-t006:** Predictive microbial clusters.

Members	*R*^2^ with Own Cluster	*R*^2^ with Next Closest	1-*R*^2^ Ratio
	Cluster 2		
*f__Victivallaceae*	0.95	0.05	0.05
*c__Alphaproteobacteria*	0.90	0.06	0.11
*f__Anaeroplasmataceae*	0.78	0.05	0.23
*o__Burkholderiales*	0.77	0.04	0.24
*k__Bacteria*	0.73	0.07	0.29
*g__Desulfovibrio*	0.71	0.10	0.33
*o__ML615J28*	0.31	0.11	0.77
*o__Bacteroidales*	0.02	0.01	0.99
	Cluster 13		
*g__Butyricicoccus*	0.66	0.05	0.36
*g__Coprobacillus*	0.62	0.004	0.38
*f__Lachnospiraceae*	0.09	0.03	0.94
	Cluster 15		
*g__Roseburia*	0.53	0.06	0.50
*g__Ruminococcus*	0.43	0.16	0.68
*g__Parabacteroides*	0.37	0.13	0.73
*g__Butyricimonas*	0.30	0.06	0.74
*g__Lactobacillus*	0.13	0.004	0.88

**Table 7 nutrients-13-00645-t007:** Multiple linear regression model statistics.

	Model *R*^2^	Sex	Cluster 15	Cluster 13	Cluster 2
SCFA	0.34	*p* = 0.0002	*p* = 0.0001	*p* = 0.0034	*p* = 0.007
Acetate	0.28	*p* = 0.011	*p* = 0.0001	*p* = 0.0042	*p* = 0.031
Propionate	0.22	*p* = 0.0002	*p* = 0.3	*p* = 0.030	*p* = 0.010
Butyrate	0.37	*p* < 0.0001	*p* = 0.0002	*p* = 0.038	*p* = 0.0042

Abbreviations: short-chain fatty acid (SCFA).

## Data Availability

Raw sequencing data is available from Qiita (https://qiita.ucsd.edu/ (accessed on 9 April 2020)), study number 13367, and the European Nucleotide Archive (https://www.ebi.ac.uk/ena/browser/home (accessed on 9 April 2020)), accession number ERP125218. Code for bioinformatic analysis and supporting data is available on GitHub (https://github.com/rlh13/Resistant-Starch (accessed on 9 April 2020)).

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
