# Peer review of "Resistant Starch Type 2 from Wheat Reduces Postprandial Glycemic Response with Concurrent Alterations in Gut Microbiota Composition"

_nutrients, 2021, doi:10.3390/nu13020645_

Round 1

Reviewer 1 Report

30 participants consumed rolls made from conventional wheat flour or wheat flour high in RS2 for 1 week each in a randomized, cross-over trial.  Stool samples for microbiome and SCFA were collected at the beginning and end of each 7-day period and serum glucose/insulin responses after a standardized breakfast containing the control or RS2 rolls were measured at the end of each period.  The results showed that after RS2 compared to control, glucose and insulin responses were lower, SCFA did not change and the microbiome was altered.  It was concluded that RS2-enriched wheat starch may improve metabolic indices and reduce risk of metabolic diseases compared to regular wheat and that the gut microbiome may play a role in mediating the individual’s metabolic response.

MAJOR CONCERNS

  1. The conclusions are not supported by the results in that: a) there is no demonstration of an improvement in metabolic indices – “improve” implies a benefit; there was a reduction in glucose and insulin response but no demonstration that such a change is beneficial. b) there was no demonstration that the microbiome plays a role in mediating the individual’s metabolic response to RS2 – what was shown was that fasting glucose concentration and glucose and inulin AUC’s were correlated with selected measures of fecal SCFA – such information is not new.  What would need to be shown to support the suggestion made is that the differences in glucose and insulin responses between the conventional and RS2 treatments were correlated with the respective changes in SCFA concentrations.  Furthermore, the only metabolic indices measured were glucose and insulin responses and the wording should indicate this.  Please avoid the use of subjective terms (eg. “improve”, “benefit”) and use objective language (eg. “reduce”, “changed”) to describe the results and formulate conclusions.
  2. The conclusions depend heavily on the glucose and insulin responses. The most important thing the reader needs to know in order to interpret these responses is how much available carbohydrate the test meals contained.  Unfortunately this cannot be determined.
  3. Nutrition information is incomplete in several ways: a) Table 2 does not indicate what the numbers represent (per 100g, per roll, how much does a roll weigh?); b) How were the values on Tables 2 and 3 obtained? c) How is “carbohydrate” defined?  Does “carbohydrate” include dietary fiber and resistant starch?  Does dietary fiber include resistant starch?  The main unknown is how much available carbohydrate the rolls and test meals contained. d) the weights of the ingredients in the test meal are not given.  Were there any other ingredients (mayonnaise, butter, lettuce, tomato, a drink, etc)? Please provide this information.
  4. Nutrition information is inconsistent because the amount of carbohydrate shown in Table 3 is inconsistent with that in Table 2. In table 2 the carbohydrate:fiber ratio for the RS roll is nearly 4, in Table 3 it is about 0.5.  Please review these tables to ensure the information presented is correct.
  5. The collection of only 1 blood sample between 0 and 2h after consuming a test-meal is insufficient to provide and accurate representation of the glucose and insulin responses; at the very least samples at 30 and 90 min are required, and preferably more frequent over the first 60 min.
  6. The glucose curves shown in the supplementary material show glucose going down after the RS meal; I would expect what I estimate to be about 40-45g of available carbohydrate to cause some rise in blood glucose, which has been missed because blood was not sampled between 0 and 60min. Another odd thing about the glucose response curves is that the average fasting glucose is shown to be about98mg/dl.  This is quite different from the mean of 91mg/dl shown on Table 1.

OTHER COMMENTS

  1. Please ensure all the tables and figures indicate what the values represent (means, SEM, SD, 95%CI, etc) and how to interpret the plots shown in Figure 3 and elsewhere.
  2. To be consistent with other literature in the area (allowing the reader to interpret the results in that context) it would be helpful to replace Figure 3 with the curves showing the glucose and insulin responses and to display AUC and peak in a table using means and SEM or medians and a range of %iles.
  3. Table 5: presumably the concentrations shown represent the overall means for the 4 fecal samples (before and after each test period). Please consider showing the means and SD for each of the 4 samples it can be indicated that none of the differences was significant.
  4. Line 22 in the Abstract. SCFA production was not measured … fecal SCFA concentration was measured – since most of the SCFA produced are absorbed from the colon, fecal concentration likely does not reflect production.

Author Response

Dear Editor,

We appreciate the reviewers taking time to evaluate our manuscript.  Our point-by-point responses are below.

Respectfully,

Riley Hughes and Mary E. Kable

Reviewer 1

MAJOR CONCERNS

  1. The conclusions are not supported by the results in that:
  1. a) there is no demonstration of an improvement in metabolic indices – “improve” implies a benefit; there was a reduction in glucose and insulin response but no demonstration that such a change is beneficial.

The word “improve” has been removed from the title and abstract in reference to observations of metabolic indices in this study. Additionally, we added clarifying detail when we were referring to conclusions regarding gut microbial fermentation or glycemic response. Specific changes are enumerated below.

  1. The title was changed to “Resistant starch type 2 from wheat reduces postprandial glycemia with concurrent alterations in gut microbiota composition”
  2. Line 26: “both improved glycemic response” was removed
  3. Lines 27-29: The concluding sentence of the abstract was altered to, “These effects show that RS2-enriched wheat consumption results in a reduction in postprandial glycemia, altered gut microbial composition and increased fermentation activity relative to wild-type wheat.”
  4. b) there was no demonstration that the microbiome plays a role in mediating the individual’s metabolic response to RS2 – what was shown was that fasting glucose concentration and glucose and inulin AUC’s were correlated with selected measures of fecal SCFA – such information is not new.  What would need to be shown to support the suggestion made is that the differences in glucose and insulin responses between the conventional and RS2 treatments were correlated with the respective changes in SCFA concentrations.  Furthermore, the only metabolic indices measured were glucose and insulin responses and the wording should indicate this.  Please avoid the use of subjective terms (eg. “improve”, “benefit”) and use objective language (eg. “reduce”, “changed”) to describe the results and formulate conclusions.
  5. Lines 22-23: has been altered to include more detail regarding the microbial fermentation response to resistant starch consumption.
  6. Lines 27-29 were altered as described above
  7. Lines 567 – 571 were removed from the discussion section and observations were clarified in lines 576-578

2. The conclusions depend heavily on the glucose and insulin responses. The most important thing the reader needs to know in order to interpret these responses is how much available carbohydrate the test meals contained.  Unfortunately this cannot be determined.

We agree that the true available carbohydrate content of the test meals and rolls cannot be determined with certainty.  For this research, an analysis of the rolls was performed by Medallion labs for soluble and insoluble fibers and resistant starch. The total carbohydrate content was determined by calculation, subtracting moisture, protein, fat, and ash content. Hence, we will label the carbohydrate as ‘estimated by calculation.’

Nutrition information is incomplete in several ways:

  1. a) Table 2 does not indicate what the numbers represent (per 100g, per roll, how much does a roll weigh?)

A footnote has been added to Table 2 to clarify that nutrition information is per 1 roll. Each roll weighed approximately 92 grams.

  1. b) How were the values on Tables 2 and 3 obtained?

The nutritional information for the rolls shown in Table 2 was obtained from Medallion labs. Detail has been added regarding the methods used to calculate carbohydrate and fiber content. The nutritional information for the test meals was determined using NDSR with the recipe for the test meal provided to NDSR. Footnotes have been added to both tables to provide relevant information.

  1. c) How is “carbohydrate” defined?  Does “carbohydrate” include dietary fiber and resistant starch?  Does dietary fiber include resistant starch?  The main unknown is how much available carbohydrate the rolls and test meals contained.

Total carbohydrate was determined by difference and is therefore an estimate. Clarification has been added to table footnotes. Total carbohydrate is approximately equal to the sum of starch, sugars, and dietary fiber. Dietary fiber values do not include resistant starch content as resistant starch could not be determined in NDSR. True available carbohydrate content of the test meals and rolls cannot be determined with certainty, and is therefore not included in the tables.

  1. d) the weights of the ingredients in the test meal are not given.  Were there any other ingredients (mayonnaise, butter, lettuce, tomato, a drink, etc)? Please provide this information.

The other ingredients in the test meals are listed on lines 147-150 (egg, cheese, turkey sausage, butter). Weights of each item have been added for clarification.

3.Nutrition information is inconsistent because the amount of carbohydrate shown in Table 3 is inconsistent with that in Table 2. In table 2 the carbohydrate:fiber ratio for the RS roll is nearly 4, in Table 3 it is about 0.5.  Please review these tables to ensure the information presented is correct.

Our apologies, the carbohydrate values have been corrected for the test meal.

4. The collection of only 1 blood sample between 0 and 2h after consuming a test-meal is insufficient to provide and accurate representation of the glucose and insulin responses; at the very least samples at 30 and 90 min are required, and preferably more frequent over the first 60 min.

The first postprandial blood sample was obtained 1 hour after the test meal was consumed, not 2 hours. It is unfortunate that peak values for both glucose and insulin may have occurred earlier, possibly around 30 minutes. This is a limitation of the study and has been added to the list of limitations in the Discussion section in lines 580-582. Even considering this limitation, the analysis of the 3 postprandial samples obtained at 1, 2 and 3 hours after the meal, still provides valuable information about glycemia and insulinemia that are important in this investigation.

5. The glucose curves shown in the supplementary material show glucose going down after the RS meal; I would expect what I estimate to be about 40-45g of available carbohydrate to cause some rise in blood glucose, which has been missed because blood was not sampled between 0 and 60min. Another odd thing about the glucose response curves is that the average fasting glucose is shown to be about98mg/dl.  This is quite different from the mean of 91mg/dl shown on Table 1.

Although the postprandial glucose response does go down in response to the RS meal, this response was consistent from subject to subject; thus we are very confident that this phenomenon is real. And yes, agreed, that any carbohydrate that was absorbed before 60 min would have been missed. The mean of 91 mg/dl was obtained from participants’ baseline screening visits while the glucose curves were obtained from their test day visits. This is what has caused the difference in the average fasting values.

OTHER COMMENTS

7. Please ensure all the tables and figures indicate what the values represent (means, SEM, SD, 95%CI, etc) and how to interpret the plots shown in Figure 3 and elsewhere.

Wording has been added to figures to explain the structure of violin plots and to Figure S1 to indicate that lines shown are mean and 95% CI. Wording in the footnotes of tables has already been added to explain values (e.g., mean + standard deviation)

8. To be consistent with other literature in the area (allowing the reader to interpret the results in that context) it would be helpful to replace Figure 3 with the curves showing the glucose and insulin responses and to display AUC and peak in a table using means and SEM or medians and a range of %iles.

The postprandial curves have been moved from the supplementary material to replace Figure 3. A table has also been added to describe the mean and standard deviation for postprandial glucose and insulin iAUC and peak.

9. Table 5: presumably the concentrations shown represent the overall means for the 4 fecal samples (before and after each test period). Please consider showing the means and SD for each of the 4 samples it can be indicated that none of the differences was significant.

A table has been added to the supplementary material to show SCFA concentrations for each treatment period.

10. Line 22 in the Abstract. SCFA production was not measured … fecal SCFA concentration was measured – since most of the SCFA produced are absorbed from the colon, fecal concentration likely does not reflect production.

Thank you for pointing out this distinction. Wording has been revised.

Reviewer 2 Report

This is a clearly designed and executed study, and the investigation of the effects of wheat derived RS2 on the gut microbiota and fermentation indices has been very well done. Kudos to the authors, I would recommend  following these minor changes:

Line 69-71 - italicize bacteria names

Line 109 - describe randomisation procedure (computer program, coin flip, etc?)

Line 213 - was 50bp trimmed off both the forward and reverse reads?

Line 215 - "average of 3512 sequences per sample". - please provide range

Line 216 - "samples with fewer than 533 samples were excluded" - how many samples were excluded?

Additionally, I would recommend making the code used in this analysis available, i.e. provide link to a github repo in the manuscript.

Author Response

Reviewer 2

This is a clearly designed and executed study, and the investigation of the effects of wheat derived RS2 on the gut microbiota and fermentation indices has been very well done. Kudos to the authors, I would recommend following these minor changes:

Line 69-71 - italicize bacteria names

Thank you for pointing this out; names have been italicized.

Line 109 - describe randomisation procedure (computer program, coin flip, etc?)

Details of the randomization procedure have been added.

Line 213 - was 50bp trimmed off both the forward and reverse reads?

Only the forward reads were used. Wording has been added to clarify.

Line 215 - "average of 3512 sequences per sample". - please provide range

The range has been added to provide more detail.

Line 216 - "samples with fewer than 533 samples were excluded" - how many samples were excluded?

Only one sample was excluded. This detail has been added.

Additionally, I would recommend making the code used in this analysis available, i.e. provide link to a github repo in the manuscript.

A GitHub repository has been created containing the code used to perform the analyses presented.  It can be found at https://github.com/rlh13/Resistant-Starch.  This information has been added to the data availability statement.

Round 2

Reviewer 1 Report

Thank you for addressing my concerns quite well.  I was not concerned that carbohydrate was measured by difference - it was that I am not sure about how much unavailable carb is present - since my understanding of the fiber analysis is that it does include some RS.

Author Response

Thank you for addressing my concerns quite well.  I was not concerned that carbohydrate was measured by difference - it was that I am not sure about how much unavailable carb is present - since my understanding of the fiber analysis is that it does include some RS.

We apologize for the confusion. You are correct that the methods used do not allow us to determine what fraction of the measured fiber is composed of resistant starch.  We have amended the footnote in Table 2 to try to further clarify the methods used for fiber analysis and to illustrate this limitation (lines 129-134).